# Spillover Risks on Cryptocurrency Markets: A Look from VAR-SVAR Granger Causality and Student's-t Copulas

**Toan Luu Duc Huynh** [1,2] 

1   School of Banking, University of Economics Ho Chi Minh City, Ho Chi Minh City 700000, Vietnam;
    toanhld@ueh.edu.vn or toan.huynh@whu.edu
2   WHU—Otto Beisheim School of Management, Burgplatz 2, D-56179 Vallendar, Germany

**Abstract:** This paper contributes a shred of quantitative evidence to the embryonic literature as well as existing empirical evidence regarding spillover risks among cryptocurrency markets. By using VAR (Vector Autoregressive Model)-SVAR (Structural Vector Autoregressive Model) Granger causality and Student's-t Copulas, we find that Ethereum is likely to be the independent coin in this market, while Bitcoin tends to be the spillover effect recipient. Our study sheds further light on investigating the contagion risks among cryptocurrencies by employing Student's-t Copulas for joint distribution. This result suggests that all coins negatively change in terms of extreme value. The investors are advised to pay more attention to 'bad news' and moving patterns in order to make timely decisions on three types (buy, hold, and sell).

**Keywords:** Bitcoin; cryptocurrency; spillover risks; Copulas; Student's-t

**JEL Classification:** C46; F38; G01

---

*When you do physics you are playing against God; in finance, you are playing against God's creatures.*

(Emanuel Derman)

## 1. Introduction

One of the financial decisions is to determine the interconnection among assets because these relationships are relative reference sources to portfolio management and hedging strategies. This matter has attracted many scholars' attention in both literature and empirical fields. Notably, in the volatile world, there are many events, which not only foster, but also restrict the financial markets, financialization economies, or financial technologies. For instance, the constraints imposed by the trade war between the United States and China, the booming of many Fintech companies, the boom and bust of the Bitcoin lifecycle, etc.

The development of cryptocurrencies is an irresistible process. With the presence of Bitcoin and the rest of coins in market (referring to Altcoins, which are established for alternative investment after the commencement of Bitcoin), many investors paid close attention to them as a potential market for earning money. Unbelievably, Bitcoin returns used to climb up by 1358% in 2017 alone (Bouri et al. 2018a). Afterwards, numerous financial institutions, such as the Chicago Mercantile Exchange (CME) Group and the Chicago Board Options Exchange's (CBOE), accepted this coin as a part of their playground in derivatives. Bitcoin became a 'financial phenomenon' due to being continuously mentioned on the Internet and social media. Suddenly, the collapse of Bitconnect caused a 'huge crash' in the total market. Many coins are likely to go "back to the drawing board" from zero. More explicitly, this shock makes the investors more afraid to invest their money.

Concomitantly, there are many controversial studies regarding economic characteristics of Bitcoin and these cryptocurrencies. It is undeniable that if there is a shock in cryptocurrency market, it might be a spillover for other coins (Huynh et al. 2018; Koutmos 2018a, 2018b). This concern is just a beginning because some studies explored the relationship between Bitcoin and other financial assets such as currencies (Dyhrberg 2016), global macroscopic determinants (Ciaian et al. 2016), macroeconomics news (Al-Khazali et al. 2018), energy assets (Bouri et al. 2017a), uncertainties (Bouri et al. 2017b), other assets (Bouri et al. 2018b), etc. However, there is a shortage gap in bridging the relationships specializing in spillover risks among cryptocurrencies.

There are also several studies that examine the contagion risks and spillover effects in cryptocurrencies, such as the studies of Fry and Cheah (2016), Yi et al. (2018), Ji et al. (2018b), Katsiampa (2018) and Catania and Sandholdt (2019), etc. However, these papers are initial exploiters of quantitative techniques for contagion risks among cryptocurrencies. In the following parts, we will discuss them to point out why we started working using the VAR-SVAR and Copulas approach. Therefore, this study contributes to the existing literature and empirical evidence different kinds of quantitative techniques that estimate the spillover risks among cryptocurrencies. To be more precise, this study has three main contributions: Firstly, in this study, we drew the updated data before and after the 'Bitcoin crash', which might capture the spillover effect clearly. Secondly, we employed a rich set of quantitative techniques including the linear approaches (Pearson correlation and VAR) and structural approaches (Structural VAR). Thirdly, we also introduce the cutting-edge viewpoint in using t-Student's Copulas, mainly based on the Student's-t distribution to estimate the dependence structure for extreme value events, which is considered our major contribution.

In this study, we attempt to employ the different econometric methods to answer the following research questions. (i) Are there spillover effects among the cryptocurrency market from empirical data? (ii) Is there any coin which is dependent on the other movements? (iii) Are there spillover effects in extreme events? In order to achieve this research contribution and to answer these research questions, we employed four main econometric models, namely Pearson correlation, VAR causality, SVAR causality, and t-Student's Copulas.

The remainder of this paper is structured as follows. Section 2 acknowledges the existing literature review regarding Bitcoin and spillover methodology measurement. Section 3 describes our data collection and briefly summarizes our methodologies. Section 4 illustrates our findings and results. Finally, Section 5 will conclude and deliver some implications.

## 2. Literature Review

In this section, thanks to the previous theoretical and empirical studies, we will review them as three categories: cryptocurrency markets, spillover risks, and relevant fields.

### 2.1. Terminology and Basic Concepts

The digital era has gradually changed monetary regimes. Regarding the terms of the 'cryptocurrency market', there is the digital exchange, which provides the platform for nonphysical coins or notes. Therefore, the exchange status exists in online form. Noticeably, the transactions have continued with the innovative terminology 'cryptography'[1]. This work should be conducted to secure the previous transactions, which are required to record as well as update the electronic ledger-known as a 'blockchain'.

'Cryptocurrency' refers to the 'digital coin' which operates in the previous aforementioned context. In this following part, we will focus on summarizing characteristics of some typical cryptocurrencies in our studies. One of the coins worth mentioning is Bitcoin, first introduced in 2009 by Satoshi Nakamoto. Especially, there were many skeptics that the Bitcoin price climbed up after its ten-year existence.

---

[1]　It becomes one of popular practices to perform the verification for historical online confirmation among trading parties.

Afterwards, Ethereum came to the market with new functions and differences to Bitcoin. For instance, Ethereum demonstrates a decentralized platform which supports smart contracts to save trading time as well as transaction cost. Therefore, technically different from Bitcoin, Ethereum allows the users to program without any possibilities of third-party intervention, fraud, etc. Meanwhile, Ripple was born for the purpose of a real-time gross settlement system. Therefore, Ripple is likely to generate the currency and remittance network, which freely enables trading transactions. In addition, Litecoin is also known as the 'peer-to-peer network'. This coin is also built on the fundamental concept of Bitcoin but it is also an open-source without any central authorization. Similarly with Ethereum, Stellar has some noticeable features such as an open-source, decentralized protocol. In addition, this coin gradually fosters the transferring process with cross-border transactions of many cryptocurrencies. In brief, there are some commonly shared technical characteristics among these cryptocurrencies, such as (i) decentralized platform, (ii) the usage of cryptography for security, (iii) anonymous transactions, and (iv) quick confirmation.

Time after time Bitcoin and its sequential coins (Ethereum, Ripple, Litecoin, Stellar, etc.) have attracted the investors' attention. Therefore, these coins have gradually become the emerging financial asset class (Brandvold et al. 2015; Polasik et al. 2015). In terms of 'cryptocurrency market', we would like to address some concerns. First, contradictory to traditional monetary regimes, these cryptocurrency prices are wholly depended on perception as well as willingness to pay by investors. It is very intuitive to consider that all investors are 'peer-to-peer' participants in the market having no central control Sovbetov (2018). Second, the majority of cryptocurrencies represents the limitation in mining or producing new ones. This means that the limited supply is one of important traits. Coin miners tend to consume large amounts of energy as well as computer memory to generate a 'reward', meaning a new coin. Thus, it is deterministic, meaning that the coin stake can be generated in this way, not depending on the wealth. Therefore, the increase in prices can be explained by the supply–demand of trading investors. Thus, the movement of this market mainly depends on investor behavior. Third, the hashes of current blocks for one coin (which mainly influences the current price) are the mathematical function deriving from Nakamoto (2008):

$$Hash\ of\ current\ block = f(\theta_{t-1},\ \phi,\ \Phi)$$

In which, $\theta$ is the hash, $\phi$ is the difficulty level of mining the coin, and $\Phi$ is the random key generated in the current process ($\Phi$ is uniquely specific for one stage). This cryptocurrency market has past memories, difficulties in market, and typical characteristics in each transaction stage.

Furthermore, based on the aforementioned characteristics, this market is quite different from the stock market. We will discuss the three distinct features of these differences. First, the cryptocurrency market is traded 24/7, while the stock market is off during holidays, Saturday, and Sunday. Second, some stock exchanges regulate the upper lower bound for trading, whereas the cryptocurrency market is free to trade. This means that the cryptocurrency has high volatility which allows for an increase in price of up to 100 times. Third, the stock has intrinsic value, which is mainly based on the operating income or financial activities. Meanwhile, the cryptocurrencies have no intrinsic or tangible value. This kind of asset is mainly based on the fundamental concept of 'hash of current block', which is aforementioned by determinants in the previous function.

### 2.2. Bitcoin and Cryptocurrency Markets

From the policy-makers' perspective, there is an opposite view to criticize cryptocurrencies, which impedes the stability and development of financial markets. It derives from the studies of Gandal et al. (2018), Lo and Wang (2014), and Velde (2013). However, the development of Bitcoin, as well as other cryptocurrencies, still happens with many warnings and restrictions.

When it comes to economic characteristics, many studies doubted whether Bitcoin is a kind of currency or not. Bouri et al. (2017b) admitted that Bitcoin is considered as a 'peer-to-peer' environment

that allows the alternative financial asset to be easily traded. However, Yermack (2015) disagreed with previous studies. This study claimed that Bitcoin should be not used as money, whereas Polasik et al. (2015) looked at Bitcoin as a mean of exchange, not reflected entirely as a monetary currency. On the other hand, Selgin (2015) supposed that Bitcoin is a kind of counterfeit commodity money.

There are also inconclusive findings regarding investors' behaviors in cryptocurrency markets, especially Bitcoin, in terms of forecasting their prices. Cheah et al. (2018) emphasized that Bitcoin shows an uncertain cointegration relationship with other assets. Mainly, this study indicated the inefficiency and informational nonhomogeneity in Bitcoin markets. These results are similar to the studies of Ciaian et al. (2016), Katsiampa (2017), and Urquhart (2017). Interestingly, there are some researches that have evaluated the microscopic structure in the Bitcoin market. In more detail, while Dyhrberg et al. (2018) initially examined the transaction costs as well as liquidity from market structures, Li and Wang (2017) conducted an empirical study to examine the technological and economic determinants of cryptocurrency exchange rates. These studies provided insights of large factors such as news public interest (Google search, Wiki views, Twitter, Facebook, and Forum), user volume trading (volume and frequency), Bitcoin amount, hash rate, difficulty in other markets (Gold and stocks), financial stress, and economy fundamentals. Moreover, Koutmos (2018b) and Wei (2018) also employed the bivariate VAR models to further investigation regarding transaction activity as well as growing microstructure in cryptocurrency markets. Also, Li and Wang (2017) pointed out the role of attention in Bitcoin returns. Nasir et al. (2019) contributed to the existing literature stating that Google search engines could be used to forecast Bitcoin returns in one-week shock. However, this element failed to explain why Bitcoin is volatile.

Regarding the interconnection between Bitcoin and other financial indices, Kristoufek (2015) used wavelet functions to prove that Chinese market indexes are likely to be a determinant of Bitcoin returns. This conclusion is also similar to the study of Bouoiyour and Selmi (2015). Meanwhile, Briere et al. (2015) figured out that Bitcoin has high volatility but it can be taken as a hedged instrument due to their relationship with conventional assets and unconventional assets. This study suggested the optimal ratio of Bitcoin in the portfolio is ~3 percent. By using the quantitative graphs, Ji et al. (2018a) confirmed that Bitcoin is useful for hedging purposes. In addition, Dyhrberg (2016) tested this characteristic of Bitcoin by univariate GARCH models (Generalized Autoregressive Conditionally Heteroscedastic models). Afterwards, this type of investment seems effective to hedge against the volatility of U.K. currency and equities. Interestingly, the following studies also supported the previous findings such as Bouri et al. (2017b), Balcilar et al. (2017), etc.

One of Bitcoin's features that has attracted many scholars' attention is 'bubble and speculation'. Cheah and Fry (2015) argued that Bitcoin is an only speculative asset because it does not convey any economic value. Moreover, Cheung et al. (2015) indicated that Bitcoin has speculative bubble features, which many investors use to exploit Bitcoin returns. This shares the same result with the study of Corbet et al. (2018).

In the scope of this study, we briefly summarize Bitcoin and cryptocurrencies studies based on five main points: (i) legal and policy-makers; (ii) currency or not; (iii) behaviors in markets; (iv) interconnection between cryptocurrency and other financial assets; and (v) speculative and bubble.

*2.3. General Spillover Risks*

First and foremost, the connectedness between decentralization policy and spillover effects is mentioned in study of Ogawa and Wildasin (2009) as well as Feder (2018). In these researches, the authors emphasized that decentralization might lead to spillovers among heterogeneous status. Second, when it comes to the decentralization in cryptocurrency market mentioned before, this means that these cryptocurrencies have a 'peer-to-peer' position in purchasing, selling, and trading without central control or concentrated authorization. Thus, the spillover effects might easily happen in terms of a decentralized context with unrestricted transactions.

The test of spillover risks is usually to test for many different stocks, equity markets, as well as different kinds of financial assets. In this subsection, we will acknowledge the previous studies regarding the extent of relevant models to our study. Primarily, there is also an empirical study regarding contagion risk through the Structural Vector Autoregression (SVAR) model. Dungey et al. (2011) asserted that the liquidity and volatility of JP Morgan indices measured by credit risk with spread[2], as well as a country risk by idiosyncratic shocks. Therefore, this study also emphasized the role of SVAR in identifying contagion risk through transmission of shocks from one country to another country. In the previous period, Diebold and Yilmaz (2008) suggested a method to measure the interdependent features among many assets. It is based on "own variance shares", which are estimated by error variances in forecasting. Latter, Yilmaz (2010) applied this method for measuring risks and volatility among East Asian stock markets. Similarly, there are plenty of works using Granger causality on the VAR basis to indicate the spillover risks such as Zhang et al. (2010). This study used Granger causality tests to indicate that change in the Dow Jones Industrial Average Index significantly causes changes in the Shanghai Stock Exchange Composite Index. This is one of the fundamental concepts for us to employ Granger causality for further investigation. Besides, the work of Shabri Abd. Majid et al. (2009) integrated the Generalized Method of Moments (GMM) in panel data to test contagion risks in 5 emerging markets among ASEAN countries. Additionally, Ding (2010) also checked the co-movements with Granger causality for the US and the Asia Pacific stock markets, while Central and Eastern Europe are the scope of study by Tudor (2011). Then, there is also a country research, employing this methodology of Vinh (2014) and Su (2017).

In order to observe the spillover risks, Christodoulakis and Satchell (2002), Engle (2002), as well as Tse and Tsui (2002) proposed the methodology of time-varying conditional correlations to examine whether the spillover risks happen in the market or not. There are many innovations from the previous traditional model. For example, Kundu and Sarkar (2016) introduced the STVAR-BTGARCH-M model for examining contagion risks. Afterward, Bouri et al. (2018a) built on previous studies, investigating the spillover risks between Bitcoin and financial assets (such as MSCI indices, commodity, energy, gold, US dollar, and US Treasury). One of the most impressive points in this study is to divide the sample into two regimes, for instance, bull and bear market. Additionally, Cong et al. (2008) employed multivariate VAR to indicate the transmission between oil price shocks and the equity market in China. Afterward, the study of Narayan and Narayan (2010) reexamined this approach for country research in Vietnam. Interestingly, Park and Ratti (2008) expanded their sample to the US and European countries by using Johnson and Juselius' cointegration tests before the empirical findings of Maghyereh and Al-Kandari (2007) by nonlinear cointegration analysis for Gulf Cooperation Council (GCC) countries. Another country research is from Hammoudeh and Aleisa (2004), who used VAR, likelihood ratio, and cointegration tests to examine the equity markets and New York Mercantile Exchange (NYMEX) oil futures.

One of the modern methodologies to measure the dependence structure is Copulas. Noticeably, in the equity market, Nguyen and Bhatti (2012) took advantage of the characteristics of marginal distribution and bivariate analysis to estimate the dependence structure, which is also known as spillover risks. In the early period, Bae et al. (2003) and Boyson et al. (2010) shed new light on estimating tail (or extreme) events of market returns as contagion risks. Later, the study of Luo et al. (2011) approached this methodology to explain how Chinese stock markets transmitted their risks to the other equity markets. Interestingly, Hiang Liow (2012) presented the results of the co-movements in securitized real estate and stock markets by Copulas. Following this work, Boubaker and Sghaier (2013) also implemented their Copulas results with portfolio management implications for the US financial assets, whereas Ghorbel and Trabelsi (2014) contributed a piece of empirical evidence for energy portfolio using Copulas. Besides, the research of Al Rahahleh and Bhatti (2017)

---

[2]   It was measured by the gap between the US industrial yields and the US Treasury bond.

demonstrated that there is a relationship between the information transmission and international stock markets in co-movements by Copulas.

To sum up, our literature review regarding spillover effects only focuses on three main points: (i) Granger causality test to estimate spillover risks; (ii) other time-varying models to examine contagion phenomenon; and (iii) the applications of Copulas in spillover risks.

### 2.4. Relevant Studies in Terms of Spillover Risks on the Cryptocurrency Markets

The study of Koutmos (2018a) investigated the relationship among cryptocurrencies by employing a multivariable vector autoregression (VAR). These findings suggest that there exists the spillover pattern in cryptocurrency markets, which represents interdependencies among these coins. However, this study only examines the covariates rather than joint distribution between each pair of cryptocurrencies. Therefore, it is encouraged to have further researches in distribution characteristics.

By using the value-at-risk and expected shortfall, the study of Gkillas and Katsiampa (2018) investigated tail behaviors among the five largest coins in the exchange market. This study shares some similarities in findings with Brauneis and Mestel (2018) regarding tail risks. Interestingly, the study of Brauneis and Mestel (2018) uses a rich set of quantitative techniques, such as the Ljung and Box (1978) test for autocorrelation, Wald and Wolfowitz (1940) test for random order, bootstrapped variance ratio by Chow and Denning (1993), etc. Nevertheless, these studies might miss capturing the tail dependence structure among these cryptocurrencies. In brief, the findings are quite consistent in that there exist spillover risks among these kinds of assets.

Interestingly, Corbet et al. (2017) examined the idiosyncratic characteristics of Bitcoin and came into conclusion that there exist the spillover effects among cryptocurrencies. However, this study applied the GARCH approach to investigate the volatility spillovers. Later, by using LASSO-VAR[3] approach, Yi et al. (2018) indicated that there is interconnection among cryptocurrencies in terms of returns and volatilities. This study failed to confirm that Bitcoin is the dominant element for this transmission. Therefore, Yi et al. (2018) confirmed that the spillover effects happen in cryptocurrency markets; however, this study does not emphasize the tail dependence structure. Recently, Bouri et al. (2018a, 2018b) applied the test from Phillips et al. (2015) called generalized sup Augmented Dickey–Fuller (GSADF) test for proving the multiple bubbles as well as co-explosivity in the cryptocurrency market. Once again, this study skipped the tail structure in capturing the extreme value, which might cause spillover risks. The study of Tu and Xue (2018) is entirely new, and employs Granger causality to check interrelationship between Bitcoin and Litecoin. However, this study limits two coins without more expansion for many coins. Finally, Huynh et al. (2018) asserted that there exist the contagion risks among cryptocurrency markets by using nonparametric (chi-plots and Kendall-plots) and parametric (Copulas with Normal, Clayton, and Gumbel) approaches. This study is one of the fundamental concepts for us to develop to use further quantitative techniques in examining the spillover risks.

Recently, Trabelsi (2018) investigated the volatility spillover effects among cryptocurrencies with a time–frequency–dynamic connectedness nature. The results are quite interesting to the readers because it contributes to the empirical evidences regarding connectedness within the cryptocurrency markets. Especially, this study also introduces the time of decomposition of the total spillover index, emphasized in 2–4 days. However, Trabelsi (2018) employed VAR methodology, which is good to forecast in linear shape rather structural dependence or complex structures of asset distribution. Although this study is a review, we would like to investigate insights by using quantitative techniques specifically structural VAR and Copulas for further estimation.

To the best of our knowledge, there was no further investigation among cryptocurrencies using Granger causality on the theme of VAR and SVAR. Furthermore, the previous studies only took normal

---

[3]　Least Absolute Shrinkage and Selection Operator and Vector Autoregressive Model.

Copulas with a normal distribution to capture the left-tail dependence and fail to explain the extreme value, which only Student's-t Copulas fits. Therefore, our research will bridge the shortage of previous studies on the three following main points. (i) Review the previous studies in cryptocurrencies in terms of spillover or contagion risks; (ii) contribute to new approach of Granger causality on the theme of VAR (or linear) and SVAR (or structural dependence); and (iii) reexamine the joint distribution between each pair of cryptocurrencies with Gaussian and Student's-t Copulas allowing to capture the extreme value.

## 3. Data and Methodology

### 3.1. Data

Our daily data covered five coins—bitcoin, ethereum, xrp, litecoin, and stellar—over the period from 8 September 2015 to 4 January 2019. The main reason to choose this period is to ensure the balanced availability of dataset without missing observations. Although there were some coins having higher market capitalization—EOS, Bitcoin Cash, etc.—we also eliminated them in our sample due to the shortage of dataset in comparison with altcoins. In addition, the market capitalization on 20 February 2019 was USD 134,776,251,887; these five coins at the same date were occupied by USD 103,483,837,488. Therefore, the percentage of our research samples is 76.78%. Next, we performed the statistics description test to initially evaluate how these variables distributions are.

As in Table 1, except for Bitcoin experiencing the left-skewed trait, all variables are skewed to the right side. Noticeably, the Ripple (XRP) has the heaviest value on the fat tail, and the others have a fat tail. Interestingly, we also figure out that the average return per day is between 0.1% and 0.4%. Also, it is seen that '*xrp*' has the highest loss value in min ($-61.627\%$) as well as the largest gain in max (102.7356%). While the Bitcoin movements attract many attentions, the other coins (ethereum, xrp, litecoin, and stellar) are likely to have extreme loss or gain values, which should be considered carefully. In this market, the transaction costs seem to be less than the other financial markets, which means that investors sell coins faster and more recklessly then they buy another coin. Hence, it might happen the spillover risk phenomenon, which needs to examine in different quantitative techniques.

**Table 1.** Descriptive statistics.

| Variable | Mean | Std. Dev. | Min | Max | Skewness | Kurtosis |
|---|---|---|---|---|---|---|
| bitcoin | 0.002163 | 0.040021 | $-0.20753$ | 0.225119 | $-0.2623099$ | 7.720781 |
| ethereum | 0.004276 | 0.068703 | $-0.31547$ | 0.412337 | 0.4963407 | 7.554288 |
| xrp | 0.003004 | 0.075708 | $-0.61627$ | 1.027356 | 2.987435 | 41.54075 |
| litecoin | 0.001711 | 0.057338 | $-0.39515$ | 0.510348 | 1.271329 | 15.6589 |
| stellar | 0.003104 | 0.083676 | $-0.36636$ | 0.723102 | 2.030118 | 18.3531 |

### 3.2. Methodology

In this paper, we employ a rich set of quantitative techniques such as Pearson correlation, VAR Granger causality, SVAR Granger causality, and Copulas with two specific kinds (Gaussian and Student's-t). Importantly, Huynh et al. (2018) indicated that there are contagion risks among cryptocurrencies by using three kinds of Copulas (Normal, Clayton, and Gumbel). This study also indicated that these pairs capture the Clayton Copulas (for the left-tail dependence). However, this study fails to evaluate the extreme values, which is one of the characteristics in financial returns data. Therefore, we further investigation on Gaussian Copulas (normally known as normal) and Student's-t Copulas (for extreme value in tail dependence) (Chen and Fan 2006). Therefore, we focused on the Gaussian and Student's-t Copulas for further investigation. Lastly, this paper also takes a closer look in this market by the Pearson, VAR, and SVAR Granger causality employed by Zhang et al. (2010), Shabri Abd. Majid et al. (2009), Ding (2010), Tudor (2011), Vinh (2014), Su (2017) for an explanation of

spillover effects, to understand which coin influences another one. We will present the basic framework of our methodologies to use in the following sections.

### 3.2.1. Pearson Correlation

This is the traditional approach to investigate the relationship between two variables. There are numerous studies regarding these parameters: Stigler (1986), Snedecor and Cochran [1980] 1989, Galton (1889), etc. In this paper, we briefly acknowledge the relevant formula as well as the confidence level for testing. The parameter $\rho$ represents the product–moment correlation coefficient.

$$\hat{\rho} = \frac{\sum_{i=1}^{n} w_i (x_i - \overline{x})(y_i - \overline{y})}{\sqrt{\sum_{i=1}^{n} w_i (x_i - \overline{x})^2} \sqrt{\sum_{i=1}^{n} w_i (y_i - \overline{y})^2}} \tag{1}$$

In which, $w_i$ denotes weight and $\overline{x}$ and $\overline{y}$ are the means of $x$ and $y$, respectively. Then, we also demonstrate the unadjusted significance level for testing significance level.

$$\rho = 2*\text{ttail}\left(n - 2,\ |\hat{\rho}|\frac{\sqrt{n - 2}}{1 - \hat{\rho}^2}\right) \tag{2}$$

One of Pearson correlation benefits is that it is easy to calculate; however, it only supports linear dependence between two variables. Therefore, we also employ further investigation.

### 3.2.2. Vector Autoregressive Model (VAR)

We refer to the studies of Lütkepohl (2005) and Greene (2008) to briefly explain the VAR model in terms of linear regression without constraint placed on the coefficients. The VAR(p) model with exogenous variables is statistically written in form as

$$y_t = AY_{t-1} + B_0\chi_t + u_t \tag{3}$$

In which $y_t$ is the matrix with $(K \times 1)$ of endogenous variables; $A$ is a matrix with $(K \times K_p)$ of coefficients of lagged values of $Y$ $(Y_{t-1})$; $B_0$ is matrix with coefficients of matrix $\chi$; $\chi_t$ is the matrix $(M \times 1)$ of exogenous variables; and $u_t$ is the matrix $(K \times 1)$ of white noise innovations. Finally, $Y_t$ is the matrix $(K_p \times 1)$ matrix with $Y_t = \begin{pmatrix} y_t \\ \vdots \\ y_{t-p+1} \end{pmatrix}$.

The matrix $\chi_t$ also includes intercept terms in VAR model. Therefore, $\chi_t$ will be empty when it includes no exogenous variables and no intercept terms in the model. In summary, VAR is a model with K variables regressed in linear functions. In this estimation, there are (p) own lagged values of variables and (p) lags of other $(K - 1)$ variables, and possibly exogenous variables. Therefore, a VAR model with p lags denotes as VAR(p).

### 3.2.3. Structural Vector Autoregressive Model (SVAR)

In this study, we briefly introduced the theoretical framework of Structural Vector Autoregressive Model (SVAR) by Amisano and Giannini (2012). Assume that we have the full system of variables of interactions as follows

$$y_t = \delta_1 + b_{12}\vartheta_1 + \gamma_{11}y_{t-1} + \gamma_{12}\vartheta_{t-1} + \varepsilon_{1t} \tag{4}$$

$$\vartheta_1 = \delta_2 + b_{21}y_1 + \gamma_{21}y_{t-1} + \gamma_{22}\vartheta_{t-1} + \varepsilon_{2t} \tag{5}$$

Therefore, we can capture the mutual interaction between these variables in our model. When it comes to matrix notation, we have

$$
\begin{bmatrix} 1 & -b_{12} \\ -b_{21} & 1 \end{bmatrix} \begin{bmatrix} y_t \\ \vartheta_1 \end{bmatrix} = \begin{bmatrix} \delta_1 \\ \delta_2 \end{bmatrix} + \begin{bmatrix} \gamma_{11} & \gamma_{12} \\ \gamma_{21} & \gamma_{22} \end{bmatrix} \begin{bmatrix} y_{t-1} \\ \vartheta_{t-1} \end{bmatrix} + \begin{bmatrix} \varepsilon_{1t} \\ \varepsilon_{2t} \end{bmatrix} \tag{6}
$$

Compactly, $\xi$, denotes the matrix in Equation (6).

$$
B\xi_t = \Delta + \Gamma\xi_{t-1} + \varepsilon_t \tag{7}
$$

This SVAR can be rewritten in a VAR form by premultiplication by $B^{-1}$

$$
\xi_t = A_0 + A_1\xi_{t-1} + u_t
$$

In which, $u_t = B^{-1}\varepsilon_t$, $A_0 = B^{-1}\Delta$, and $A_1 = B^{-1}\Gamma$, then the $\varepsilon_t$ is assumed that SVAR generates white noise and i.i.d, the $u_t$ from VAR has the following characteristics; (i) zero mean, (ii) fixed variance, (iii) not individually autocorrelated and, most importantly, (iv) the correlation between $b_{12}$ and $b_{21}$ is different to 0. Finally, our calculation will examine: $u_t = B^{-1}\varepsilon_t = \begin{cases} \frac{(\varepsilon_{1t}-b_{12}\varepsilon_{2t})}{1-b_{12}b_{21}} \\ \frac{(\varepsilon_{2t}-b_{21}\varepsilon_{1t})}{1-b_{12}b_{21}} \end{cases}$.

There are also two types of SVAR estimation such as short-run and long-run restrictions. For the short run one, the Cholesky decomposition is followed by a study of Sims (1980), whereas Amisano and Giannini (2012) adapted long run SVAR to the methodology of checking local identification.

### 3.2.4. Granger Causality

One variable, $x$, is defined as having Granger causality on variable $y$ if the given previous information of $y$, as well as the past values of $x$, enable to forecast the current value of $y$. This is the fundamental concept of Granger causality of these variables. Therefore, the null hypothesis is to test that the joint coefficients equal zero, including the lagged values of $x$ in the regression between $y$, the lagged values of $y$ and $x$. Therefore, Granger (1969) employed the Wald statistics for the hypothesis. Especially, the procedure for the Granger causality test is to store all values in VAR (or SVAR) regression first. Afterwards, they calculate and report small sample F statistics or large sample $\chi^2$ statistics for the null hypotheses.

### 3.2.5. Copulas Approaches

According to Gudendorf and Segers (2010), Copulas refers to specific multivariate distribution functions, in which the distribution function $H$ of a n-dimensional random vector $X = (X_1, \ldots, X_d)$ is the function defined by

$$
H(x) = P(X \le x) = P(X_1 \le x_1, \ldots, X_d \le x_d) \tag{8}
$$

with $x = (x_1, \ldots, x_d) \in \mathbb{R}^d$.

The distribution function $F_j$ of $X_j$ with $j \in \{1, \ldots, d\}$ can be recalled by the multivariate distribution function $H$ by $F_j(x_j) = H(\infty, \ldots, \infty, x_j, \infty, \ldots, \infty)$, $x_j \in \mathbb{R}$. Therefore, $F_j, \ldots, F_d$ is also known as univariate margins of $H$ (or called as marginal distribution functions of $X$). One of the concise Copulas definitions is a multivariate distribution function with standard uniform univariate margins, that is, U(0, 1) margins.

- *Gaussian Copulas*

Malevergne and Sornette (2006) also asserted that the parameter for dependence structure $R_g$ can be estimated as follows

$$\hat{R}_g = \underset{R_g}{\text{argmax}} \sum_{i=1}^{n} \log c_g(\widetilde{x}_i; R_g) \tag{9}$$

In which, $\widetilde{x}$ is considered as increasing transformations of variable $x$. Then, random vector $\widetilde{x}$ and $x$ share the same Copulas function. The method of maximum likelihood estimates this parameter in Equation (5). To be more specific, $c_g(\widetilde{x}, R_g)$ denotes the density function of the Gaussian Copulas with parameter $R_g$.

- *Student's-t Copulas*

The parameter of dependence structure $R_t$ and the degree of freedom $v$ can be estimated by maximum likelihood:

$$(\hat{R}_t; \hat{v}) = \underset{(R_t; v)}{\text{argmax}} \sum_{i=1}^{n} \log c_t(\widetilde{x}_i; R_t; v) \tag{10}$$

In which, $c_t(\widetilde{x}, R_t; v)$ denotes the density function of the Student's-t Copulas with parameter $(R_t; v)$. This refers to the study of Kotz and Nadarajah (2004).

## 4. Empirical Findings

### 4.1. Correlation Matrix

One of the basic analyses from these coins' return is to test these correlations. We perform the Pearson method (Pearson 1896) for estimating the product–moment correlation coefficient with testing under unadjusted significance level. There is a correlation matrix after testing at this significance level.

As seen in Table 2, all pairs of coins have a strong correlation at the 1% significance level. It displays that these coins returns have the dependence of linearity. This is the initial stage to investigate their structural dependence. This suggests that any positive change at one coin return might lead to positive change in other coins' returns (ethereum, xrp, litecoin, and stellar) due to the positive signs. However, we also need the further quantitative investigation to conclude the spillover phenomenon in this market.

**Table 2.** Correlation matrix.

|          | Bitcoin     | Ethereum    | xrp         | Litecoin    | Stellar |
|----------|-------------|-------------|-------------|-------------|---------|
| bitcoin  | 1           |             |             |             |         |
| ethereum | 0.3992 ***  | 1           |             |             |         |
| xrp      | 0.3043 ***  | 0.2587 ***  | 1           |             |         |
| litecoin | 0.6113 ***  | 0.3871 ***  | 0.3609 ***  | 1           |         |
| stellar  | 0.3661 ***  | 0.2789 ***  | 0.5488 ***  | 0.3857 ***  | 1       |

The symbols *, **, and *** denote the significance at the 10%, 5%, and 1% levels, respectively.

### 4.2. Test of Stationary

In order to ensure that our further findings and results are unbiased and not spurious, we tested the stationary of our variables first. For our test, we employed ADF (Dickey and Fuller 1979) and PP (Phillips and Perron 1988) unit root tests in the initial stage. In addition, we also further employed Zivot and Andrews (2002) test for Structural Break of Stationary to our variables. Table 3 will represent the results of test of stationary for these variables.

**Table 3.** Test of stationary.

| Variables | Augmented Dickey–Fuller | Phillips–Perron | Zivot–Andrews |
|:---:|:---:|:---:|:---:|
| bitcoin | −34.983 *** | −35.005 *** | −13.900 *** |
| ethereum | −33.161 *** | −33.288 *** | −18.966 *** |
| xrp | −35.585 *** | −35.934 *** | −13.027 *** |
| litecoin | −34.731 *** | −34.809 *** | −12.725 *** |
| stellar | −32.703 *** | −32.760 *** | −14.925 *** |

The symbols *, **, and *** denote the significance at the 10%, 5%, and 1% levels, respectively.

We find that all variables are stationary at the 1% significance level. This indicates that our variables are integrated at I(0), which allows us to conduct further empirical estimation. Then, we also perform the Lütkepohl (2005)[4] test to choose the most appropriate lag order. We refer to the study of Ivanov and Kilian (2001) to choose the best-fit lag order (current value) for our Granger causality models.

*4.3. VAR Granger Causality Findings*

One variable (origin) has VAR Granger causality with another variable (receiver) when the past and current information of the origin can be statistically used to predict for the receiver. As the results stated in Table 4, there are some contagion effects among cryptocurrency markets. There is no contagion effect from the other coins on Bitcoin or Ethereum. Meanwhile, xrp is likely to bear spillover risk from bitcoin, litecoin, and stellar at significance levels of 1%, 10%, and 5%, respectively. There is weak evidence that Stellar influences on Litecoin in the Granger causality at 10% significance level. In addition, xrp significantly causes in Granger relationship to stellar at 1% level. To be more specific, there is a considerably conclusive point on bidirectional Granger causality. The only pair of two coins—'*xrp-stellar*'—shows the bilateral relationship on Granger causality test.

However, the Granger causality test introduces the spillover effect among these cryptocurrencies, but it fails to indicate the risk structure in each distribution of these coins. This is also a drawback of Granger causality test methodology. Therefore, we suggest further quantitative techniques capture the dependence structure among these coins. Table 4 will demonstrate the findings in VAR Granger causality test.

**Table 4.** VAR Granger causality results[5].

| Receiver | Origin | | | | |
|:---:|:---:|:---:|:---:|:---:|:---:|
| | **Bitcoin** | **Ethereum** | **xrp** | **Litecoin** | **Stellar** |
| bitcoin | | 0.55163 | 4.489 | 1.7486 | 5.5009 |
| ethereum | 0.14445 | | 0.74868 | 0.44237 | 1.7673 |
| xrp | 9.6048 *** | 2.4533 | | 8.5096 ** | 5.3684 * |
| litecoin | 1.0108 | 0.2165 | 2.0288 | | 5.0043 * |
| stellar | 1.359 | 1.5335 | 29.199 *** | 3.2276 | |

The symbols *, **, and *** denote the significance at the 10%, 5%, and 1% levels, respectively.

---

[4]   Selection-order criteria based on rich set of parameters such as the Log-Likelihood (LL), the Likelihood ratio (LR), the Prediction Error (FPE), the Akaike's Information Criterion (AIC), the Schwarz's Bayesian Information Criterion (SBIC), and the Hannan and Quinn Information Criterion (HQIC).

[5]   The VAR model estimation for VAR Granger causality is based on the suggested lag by Lütkepohl (2005) with the L(3) term and we also employed the multivariate VAR (all of the cryptocurrencies) in our models to test the spillover effects rather than bivariable (it might be omitted the effects from other cryptocurrencies).

### 4.4. SVAR Granger Causality Results

In the previous part, we employ VAR for Granger causality to estimate the relationship among these cryptocurrencies. However, after the crash of Bitcoin in 2017, many coins plunged. Afterward, they gradually covered. Therefore, we employed Structural Vector Autoregression for Granger causality estimation. The more advantageously quantitative results are (i) capturing the structural break of the dataset and (ii) dynamic stochastic models with a minimum of identifying assumptions. Table 5 will summarize the results of Structural Vector Autoregression (SVAR) Granger causality test.

**Table 5.** Structural Vector Autoregression (SVAR) Granger causality results.

| Receiver | Origin | | | | |
|---|---|---|---|---|---|
| | **Bitcoin** | **Ethereum** | **xrp** | **Litecoin** | **Stellar** |
| bitcoin | | 0.8461 | 6.029 *** | 0.71176 | 6.6254 *** |
| ethereum | 044158 | | 1.792 | 0.37347 | 2.2796 |
| xrp | 0.26372 | 1.7793 | | 3.1169 * | 3.3771 |
| litecoin | 0.58286 | 0.22677 | 0.06153 | | 1.8682 |
| stellar | 1.6201 | 1.6 | 16.003 *** | 3.528 ** | |

The symbols *, **, and *** denote the significance at the 10%, 5%, and 1% levels, respectively.

Interestingly, the results are similar to VAR Granger causality in terms of two pars '*xrp-stellar*' and '*litecoin-stellar*'. In which, the changes of xrp and litecoin are likely to cause the changes of stellar at significance levels of 1% and 5%, respectively. When it comes to structural break, bitcoin seems to be sensitive. Both xrp and stellar cause a change in Bitcoin at a significance level of 1%. There is also weak evidence that litecoin influences xrp at 10% significance level. However, in the SVAR Granger causality context, there is not the existence of a bidirectional relationship among these cryptocurrencies. Moreover, the differences between VAR Granger causality and SVAR Granger causality findings suggest having further quantitative techniques to investigate the dependence structure among these variables.

### 4.5. Copulas Approach

In this part, we performed the Copulas approach to test whether these pairs of cryptocurrencies have a dependence structure at the tail or not. Hence, our work tests two types of Copulas—Frank (normal) and t-Copulas (t-student Copulas)—for the dependence structure of these cryptocurrencies. Before choosing the appropriate Copulas, we need to test the dependence structure first. The Kendall parameter is one of our estimations for statistical evaluation.

As results are represented in Table 6, it can be seen that all cryptocurrency pairs have strong dependence structure through Kendall tau ($\tau$) parameters at the 1% significance level. Therefore, we employed further Copulas estimation, which detects an interpretable dependence structure for these cryptocurrencies returns. Malevergne and Sornette (2003) also indicated that the Copulas approaches (including Gaussian and the Student's-t) are taken into consideration for testing correlation in terms of structural dependence among currencies. The Tables 6 and 7 will demonstrate the results of Kendall parameters as well as Copulas test, respectively.

**Table 6.** Kendall τ parameter for dependence structure.

|          | **Bitcoin**  | **Ethereum** | **xrp**     | **Litecoin** | **Stellar** |
|----------|--------------|--------------|-------------|--------------|-------------|
| bitcoin  |              |              |             |              |             |
| ethereum | 0.282 ***    |              |             |              |             |
| xrp      | 0.287 ***    | 0.288 ***    |             |              |             |
| litecoin | 0.523 ***    | 0.316 ***    | 0.356 ***   |              |             |
| stellar  | 0.283 ***    | 0.280 ***    | 0.409 ***   | 0.336 ***    |             |

The symbols *, **, and *** denote the significance at the 10%, 5%, and 1% levels, respectively.

**Table 7.** Gaussian Copula and Student's-t Copulas estimation.

| **Pairs**          | **Gaussian Copula** | **Student's-t Copulas** |
|--------------------|---------------------|-------------------------|
| bitcoin-ethereum   | 0.4148 [115.7]      | 0.4334 **[160.1]**      |
| bitcoin-xrp        | 0.4135 [114.9]      | 0.4389 **[162.6]**      |
| bitcoin-litecoin   | 0.6894 [397.5]      | 0.7367 **[525.5]**      |
| bitcoin-stellar    | 0.4217 [120]        | 0.4328 **[161.8]**      |
| ethereum-xrp       | 0.3958 [104.4]      | 0.4467 **[145.5]**      |
| ethereum-litecoin  | 0.4484 [137.7]      | 0.4793 **[173.5]**      |
| ethereum-stellar   | 0.3842 [97.8]       | 0.4284 **[132.4]**      |
| xrp-litecoin       | 0.4872 [166.3]      | 0.5453 **[268.4]**      |
| xrp-stellar        | 0.5921 [265.5]      | 0.6001 **[330.2]**      |
| litecoin-stellar   | 0.481 [161.5]       | 0.5058 **[207.6]**      |

The maximized log-likelihood of the corresponding coefficients $\rho_0$ is shown in square brackets.

Our findings demonstrated that these pairs of cryptocurrencies have strong dependence structure on Student's-t Copulas, which should be preferred to the Gaussian one. Based on maximized log-likelihood results, we came to the conclusion that the Student's-t is a better fit for our data than its counterpart. As many previous studies evaluated that the Student's-t Copulas offers deep insights in interpreting asymptotic dependence in the tail. Therefore, it is clear to witness that these pairs of cryptocurrencies joint symmetric tail dependence. Moreover, it also represents that the spillover risks happen among these cryptocurrencies through joint fat tails mechanism (at Student's-t Copulas estimation). To be more precise, the network of contagion risks among these cryptocurrencies (including bitcoin, ethereum, xrp, litecoin, and stellar) increased the probability of joint extreme values. Clearly, this approach offers us the precise nature of correlation among these kinds of coins in terms of structural distribution. Regarding the goodness-of-fit for Copulas, Embrechts (2009) also pointed out that up to 99.9% of Copulas approach will pass through the goodness-of-fit. Therefore, we focused on choosing the Copulas family for parametric figures as Genest et al. (1995) suggest.

*4.6. Summary of Findings*

Before stating conclusions and implications, we will summarize our findings for the estimation above. Firstly, the Pearson correlation showed strong evidence about each pair of cryptocurrencies at the 1% significance level. One of significant findings to point out is that all cryptocurrencies have linear correlations. However, this approach fails to explain the dependence structure as well as extreme value. Our findings are similar to the study of Brauneis and Mestel (2018). Secondly, VAR and SVAR Granger causalities demonstrated inconsistent findings. Regarding the linear dependence through VAR estimation, ethereum is the only element that has an independent feature. This coin does not cause or receive any effect. Meanwhile, all the remaining cryptocurrencies cause spillover for another one. Especially, Ripple (xrp) is quite sensitive to changes because it receives all effects from these cryptocurrencies. When it comes to structural dependence, Bitcoin is likely to incur spillover effects, whereas Ripple (xrp), Litcoin, and Stellar tend to cause the change of the other cryptocurrencies. Our findings are quite similar to Tu and Xue (2018) regarding spillover effects. However, we cannot conclude that Bitcoin dominates the market to cause contagion risks. Thirdly, by using further investigation in quantitative techniques as Gaussian and Student's-t Copulas, we figured out that all cryptocurrencies have strong evidence independence structure. Previously, the findings of Huynh et al. (2018), who employed the three kinds of Copulas—Normal, Clayton, and Gumbel—are consistent to choose left-tail dependence structure. Our results contribute to the investigation of Student's-t Copulas in terms of spillover risks with extreme value to the existing literature. Clearly, the cryptocurrency markets with all coins have spillover effects in the structural tail dependence context with an extreme value. Therefore, when a market event happens, it might cause a downside trend for these cryptocurrencies at the same time. This quantitative result demonstrates that there is a contagion risk among the cryptocurrency markets as regards to the extreme value when using Student's-t Copulas for testing.

## 5. Conclusions and Implications

This paper sheds further light on investigating the spillover effects in the cryptocurrency market by various quantitative techniques such as VAR and SVAR Granger causality as well as Copulas with types of Gaussian and Student's-t. We found that the Ethereum is likely to be an independent relationship compared to the other coins from our sample in VAR and SVAR Granger causality approaches. Previously, when using the same methodology, Bitcoin tends to have a sensitive recipient, influenced by the other coins (ethereum, xrp, litecoin, and stellar). This suggests that the investors can use Ethereum for portfolio diversification or a hedging instrument in this kind of market. Nevertheless, we would advise caution when choosing Bitcoin as one of the investment types. The Student's-t Copulas, once again, implicates that all cryptocurrencies have joint distribution in extreme value, which might cause simultaneous downside trend with 'bad news'. Therefore, the investors or portfolio managers should pay more attention to their moving patterns as well as information in order to immediately take any actions (if needed).

Based on the previous findings and results, we make some recommendations for investors as follows. First, investors should be pay attention to moving signals in the markets. This means that any current and past changes in one coin might negatively cause the movement of the other coins. Second, Ethereum is likely to be independent part in this market. Therefore, investors are able to diversify their portfolios by adding Ethereum as a hedging instrument. Third, when extreme value happens, the investors especially concern the left-tail movement, which means the spillover risks phenomenon. These suggestions are also our contributions to current literature novelty and answers for the research questions, which are mentioned in the beginning of this paper.

However, there are some limitations of the VAR-SVAR Granger causality and Student's-t Copulas methods. When it comes to VAR-SVAR Granger causality, one of disadvantages of VAR-SVAR causality is unknown asymptotic distribution of the parameters. In addition, the hypothesis is only constructed under several restricted assumptions. Finally, this approach is not able to measure nonlinear functions

of parameters of the model, such as the exponential, inversed functions ([Droumaguet et al. 2015](#)). Therefore, we also suggest the need for further research in integrating more econometric techniques into VAR-SVAR causality such as Posterior Odds Ratio or Bayesian analysis. Regarding t-Student's Copulas limitations, although the t-Student's Copulas is better than the Gaussian Copulas, it skips the serial dependence error, which will be corrected by Value-at-Risk (VaR) and Extreme-Value-Theory (EVT). Furthermore, the t-Student's Copulas does not capture the time varying changes in dependence structure. Hence, we suggest the time-varying Copulas to fix this error.

Our research also has some limitations for further researches, which are possible to outperform. First, this study only examines the whole sample without dividing into two subsamples (before and after the crash) to estimate the spillover effects. Second, there are some further Copulas for estimation such as time-varying (which estimates one dependence structure parameter for each period) or other bivariate Copulas such as Ali-Mikhail-Haq, Joe, etc.). Third, this study does not capture the spatial spillover effects, which asserts that a shock in Bitcoin (or other coins) is more likely to affect the neighboring countries than countries which are far. We also suggest the further research in DCC-GARCH integrated Bayesian or Markov-Switching to measure this purpose. Lastly, one of the practical applications for using Copulas is to construct portfolio optimization. We suggest further study to indicate the proportions of each coin for investors to put their money in.

**Funding:** This research was funded by School of Banking, University of Economics Ho Chi Minh City (Vietnam) and Chair of Behavioral Finance, WHU—Otto Beisheim School of Management (Germany).

**Acknowledgments:** We are grateful for the anonymous referees and guest editors for their remarks. Any remaining errors are my own responsibilities. The author thanks Duy Duong for excellent research assistance.

**Conflicts of Interest:** The author declares no conflict of interest.

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
