# Peer review of "Spillover Risks on Cryptocurrency Markets: A Look from VAR-SVAR Granger Causality and Student’s-t Copulas"

_jrfm, doi:10.3390/jrfm12020052_

Round 1
Reviewer 1 Report
Some contradictory statements should be rewrittern: Abstract states that the literature on this topic is embryonic. However, on page 2 author states that "This concern is not new because..." - regarding spillovers between cryptocurrencies and "There are also many studies" (same page 2). So, is embryonic or not? It seems not due to many literature author included in the overview.
Two sentences one after another start with Meanwhile (page 2). This should be reparaphrased.
Source should be added on page 2, where author states that Bitcoin has climbed up 1358%.
Introduction should emphasize more the contribution of this paper, due to many previous existing ones. Author describes the previous literature, with many different approaches of examining the same thing on same currencies. So, what is novelty here?
Other papers should be included as Trabelsi (2018) Are There Any Volatility Spill-Over Effects among Cryptocurrencies andWidely Traded Asset Classes?
The first table on page 9 should be numerated table 1. And others should be corrected accordingly. Moreover, do not write the table above, as on page 13. In text, author should correct to e.g. "in table 2"...
Page 10, formula (2), what is ttail?
The multiplication should be something else, not x, as author uses x as well for denoting the matrix.
In empirical part of VAR modelling, which lag was used? I did not find that information. Are the VAR model results on page 15 result of bivariate models or were all cryptocurrencies included in one model? This should be clearly stated.
Recommendations for (potential) investors should be extended upon, and the contributions should be added in conclusion as well.
Author Response
SUMMARY OF REVISIONS
Title paper: Spillover risks on cryptocurrencies market:
A look from VAR-SVAR Granger causality and Student-t Copulas
Number of manuscript: jrfm-465848
We would like to express our deepest gratitude for anonymous referees for their kind review of our paper and very useful remarks and feedback, which helped us to revisit/revise our work and led to significant improvement. The summary of the actions taken in the light of referee’s remarks/feedback is as follow:
Reviewer 1
# | Reviewer comment | Replies and Action Taken | Page |
1 | Some contradictory statements should be rewrittern: Abstract states that the literature on this topic is embryonic. However, on page 2 author states that “This concern is not new because...” - regarding spillovers between cryptocurrencies and “There are also many studies” (same page 2). So, is embryonic or not? It seems not due to many literature author included in the overview. | We am grateful for your comment and We rewrote these sentences to consistent content, which you suggest revising. We have corrected this mistake, thanks for rectification. | Page 2 and Page 3 |
2 | Two sentences one after another start with Meanwhile (page 2). This should be reparaphrased. | Thank you for your comment. We reparapharase our sentences to avoid repetition. | Page 2 |
3 | Source should be added on page 2, where author states that Bitcoin has climbed up 1358%. | We thank the reviewer for suggesting to add the source for these figures. | Page 2 |
4 | Introduction should emphasize more the contribution of this paper, due to many previous existing ones. Author describes the previous literature, with many different approaches of examining the same thing on same currencies. So, what is novelty here? | Thank you for your comment. We also supplement our contributions to the current novelty in the introduction. It might be supportive part to clearly explain our contributions. | Page 3 |
5 | Other papers should be included as Trabelsi (2018) Are There Any Volatility Spill-Over Effects among Cryptocurrencies and Widely Traded Asset Classes? | We deeply thank for your suggestion. We emphasize and acknowledge the previous work from Trabelsi (2018). | Page 8 |
6 | The first table on page 9 should be numerated table 1. And others should be corrected accordingly. Moreover, do not write the table above, as on page 13. In text, author should correct to e.g. "in table 2"... | Thank you for your dedication. We changed the name of table and revised the whole paper without “the table above/the table below” to avoid the confusion to readership. It was a good idea and we have revised the paper by numbering them again in each section. Please refer to our revised paper. | All the tables and paragraphs with tables |
7 | Page 10, formula (2), what is t-tail? | Thank you for your comment. We also clarify our words by endnote in the same page for clearly reading. | Page 10 |
8 | The multiplication should be something else, not x, as author uses x as well for denoting the matrix. | Thank you for your comment. I changed the denotation for avoiding the misunderstanding of our symbol | Page 12 |
9 | In empirical part of VAR modelling, which lag was used? I did not find that information. Are the VAR model results on page 15 result of bivariate models or were all cryptocurrencies included in one model? This should be clearly stated. | Thank you very much for your suggestion. We also documented two main things (the lagged value used and the multivariate VAR with all cryptocurrencies inputted in models to estimate the spillover effects among them). | Page 15 |
10 | Recommendations for (potential) investors should be extended upon, and the contributions should be added in conclusion as well. | Thank you for your comment. We also revised and clarify in the conclusions of our paper with the recommendations for investors. | Page 23 |
Reviewer 2 Report
Using from VAR-SVAR Granger causality and Student-t Copulas, this paper examines spillover risks on cryptocurrencies market. The author argues that ‘Etherum is likely to be the independent coin in the cryptocurrency market, while Bitcoin tends to be a spillover effect recipient’.
Although this is a very interesting topic, I think that it is not a well written paper, there is no a clear structure of the paper and there is no cohesion within paragraphs. It is a very confusing paper, especially for the readers who are not familiar with the cryptocurrency market literature and its relative key terms and concepts such as Etherum and Bitcoin. Nevertheless, I think that there is plenty of potential here to make a very good contribution to the literature, but at present the article is not ready for publication.
The following are comments and suggestions that should be addressed in any revised version, together with those comments other referees may have.
1. In the introduction there could also be a clearer statement on the relevance of the present paper, what is the justification of the present paper; what is the research question of the paper? How can the paper contribute to the literature?
2. The author should clarify the concept of ‘cryptocurrency market’ and the characteristics of this market. The author should also describe the differences between the stock market and the cryptocurrency market.
3. In the paper, the author refers either to ‘Etherum’ (e.g. in the abstract) or to ‘Ethereum’ (e.g. page 8). This is confusing. Is there any difference between these concepts? The author never clarifies these concepts. What is the ‘Etherum’ and/or ‘Ethereum’? I think that ‘Ethereum’ is not a cryptocurrency and it should not be confused with ‘Ether’.
4. What is the ‘Bitcoin’? What are the main differences between ‘Bitcoin’ and ‘Etherum’/‘Ethereum’?
5. A main characteristic of the cryptocurrencies is that they are decentralized systems based on block chain technology. But, what are the implications of this (de)centralised systems for the spillover effects?
6. The author very often refers to ‘other coins’ (e.g. page 2). He/she should clarify which coins he/she refers to.
7. The author should clarify the differences between currency and bitcoins
8. The author could explore spatial spillover effects, e.g. whether a shock in Japan is more likely to affect the neighbouring countries than countries which are far.
9. What are the differences between ‘bitcoin’, ‘ethereum’, ‘xrp’, ‘litecoin’ and ‘steller’?
10. The author should also discuss the limitations of the VAR-SVAR Granger causality and Student-t Copulas methods.
Author Response
SUMMARY OF REVISIONS
Title paper: Spillover risks on cryptocurrencies market:
A look from VAR-SVAR Granger causality and Student-t Copulas
Number of manuscript: jrfm-465848
We would like to express our deepest gratitude for anonymous referees for their kind review of our paper and very useful remarks and feedback, which helped us to revisit/revise our work and led to significant improvement. The summary of the actions taken in the light of referee’s remarks/feedback is as follow:
Reviewer 2
# | Reviewer comment | Replies and Action Taken | Page |
1 | Using from VAR-SVAR Granger causality and Student-t Copulas, this paper examines spillover risks on cryptocurrencies market. The author argues that ‘Etherum is likely to be the independent coin in the cryptocurrency market, while Bitcoin tends to be a spillover effect recipient’. | We are thankful for the kind summary. | Not Applicable |
2 | Although this is a very interesting topic, I think that it is not a well written paper, there is no a clear structure of the paper and there is no cohesion within paragraphs. It is a very confusing paper, especially for the readers who are not familiar with the cryptocurrency market literature and its relative key terms and concepts such as Etherum and Bitcoin. Nevertheless, I think that there is plenty of potential here to make a very good contribution to the literature, but at present the article is not ready for publication. | We are deeply grateful for your comments. We revised our papers by adding new terminology and definition for providing the basic concepts to readers. | Page 3 & 4 |
3 | In the introduction there could also be a clearer statement on the relevance of the present paper, what is the justification of the present paper; what is the research question of the paper? How can the paper contribute to the literature? | Thank you so much for your comment. We also revised to clarify more contributions, research questions as well as methodologies to cope with our research purpose. | Page 3 |
4 | The author should clarify the concept of ‘cryptocurrency market’ and the characteristics of this market. The author should also describe the differences between the stock market and the cryptocurrency market. | We are grateful for your comment. We also documents to explain more the ‘cryptocurrency market’ as well as some characteristics. | |
5 | In the paper, the author refers either to ‘Etherum’ (e.g. in the abstract) or to ‘Ethereum’ (e.g. page 8). This is confusing. Is there any difference between these concepts? The author never clarifies these concepts. What is the ‘Etherum’ and/or ‘Ethereum’? I think that ‘Ethereum’ is not a cryptocurrency and it should not be confused with ‘Ether’. | Thank you very much for your feedback. We have corrected this mistake, thanks for rectification. Afterwards, we also corrected all things with the form of ‘Ethereum’ in order to maintain the consistency in the whole paper. | All paper |
6 | What is the ‘Bitcoin’? What are the main differences between ‘Bitcoin’ and ‘Etherum’/‘Ethereum’? | Thank you for your comments. We also documents the definition, the characteristics as well as the differences among our choices in cryptocurrencies. | Page 3,4 and 5 |
7 | A main characteristic of the cryptocurrencies is that they are decentralized systems based on block chain technology. But, what are the implications of this (de)centralised systems for the spillover effects? | Thank you for your comments. We also explain the linkage between the decentralization and spillover based on theoretical framework. | Page 7 and 8 |
8 | The author very often refers to ‘other coins’ (e.g. page 2). He/she should clarify which coins he/she refers to. | Thank you very much for your suggestion. I corrected to the rest of coins in cryptocurrency market (known as Altcoin) | |
9 | The author should clarify the differences between currency and bitcoins | We are grateful for your suggestion. We also revised our manuscript to point out differences between currency and bitcoin. | Page 3 and 4 |
10 | The author could explore spatial spillover effects, e.g. whether a shock in Japan is more likely to affect the neighbouring countries than countries which are far. | Thank you for your comments. We have to admit that this is our limitation. This means that the VAR-SVAR and t-Copulas does not capture the spatial spillover effects. We mentioned our limitation in spatial spillover effects. | Page 23 |
11 | What are the differences between ‘bitcoin’, ‘ethereum’, ‘xrp’, ‘litecoin’ and ‘steller’? | We are grateful for your suggestion. We carefully documented in our manuscript as your instructions | Page 3, 4 and 5 |
12 | The author should also discuss the limitations of the VAR-SVAR Granger causality and Student-t Copulas methods. | Thank you for your suggestion. We also discuss and mention the limitations of these methodologies. In addition, we suggest the further research. | Page 22 |
Reviewer 3 Report
Although there are other studies on the contagion risks and spillover effects in cryptocurrencies I still believe that this study adds value to the existing literature and empirical evidence quantitative techniques in estimating the spillover risks among cryptocurrencies.
This paper sheds more light on the spillover effects among cryptocurrencies market by various quantitative techniques such as VAR and SVAR Granger causality as well as Copulas with types of Gaussian and Student-t. Also it was highlighted that the Ethereum has an independent relationship from other coins in VAR and SVAR Granger causality approaches. moreover, they note that Bitcoin, however, tends to have a sensitive recipient, influenced by the other coins. suggesting that the investors can use Ethereum for portfolio
diversification or hedging instrument
The structure is fine and coherent, however the English sentence structure although understandable needs to be improved.
Author Response
SUMMARY OF REVISIONS
Title paper: Spillover risks on cryptocurrencies market:
A look from VAR-SVAR Granger causality and Student-t Copulas
Number of manuscript: jrfm-465848
We would like to express our deepest gratitude for anonymous referees for their kind review of our paper and very useful remarks and feedback, which helped us to revisit/revise our work and led to significant improvement. The summary of the actions taken in the light of referee’s remarks/feedback is as follow:
Reviewer 3
# | Reviewer comment | Replies and Action Taken | Page |
1 | Although there are other studies on the contagion risks and spillover effects in cryptocurrencies I still believe that this study adds value to the existing literature and empirical evidence quantitative techniques in estimating the spillover risks among cryptocurrencies. | We are thankful for the kind comments and encouragement. | Not Applicable |
2 | This paper sheds more light on the spillover effects among cryptocurrencies market by various quantitative techniques such as VAR and SVAR Granger causality as well as Copulas with types of Gaussian and Student-t. Also it was highlighted that the Ethereum has an independent relationship from other coins in VAR and SVAR Granger causality approaches. moreover, they note that Bitcoin, however, tends to have a sensitive recipient, influenced by the other coins. suggesting that the investors can use Ethereum for portfolio diversification or hedging instrument | Thank you very much for kind summary of our results | Not Applicable |
3 | The structure is fine and coherent, however the English sentence structure although understandable needs to be improved. | We have checked the manuscript and performed a thorough round of proofreading. | Not Applicable |
Round 2
Reviewer 1 Report
just this part was not corrected: "The multiplication should be something else, not x, as author uses x as well for denoting the matrix"; see page 14/30 so please correct this
Author Response
Dear referee,
We would like to express our deepest gratitude for anonymous referees for their kind review of our paper and very useful remarks and feedback, which helped us to revisit/revise our work and led to significant improvement. I also corrected the minor mistake in the page 14.
Once again thank you for your comment. Kind regards,
Reviewer 2 Report
To the best of knowledge, I think that the paper is ready for publication.
Author Response
We would like to express our deepest gratitude for anonymous referees for their kind review of our paper and very useful remarks and feedback, which helped us to revisit/revise our work and led to significant improvement.
Thank you very much again. Kind regards.